# 1 A Reanalysis-Based Global Tropical Cyclone Tracks

# 2 Dataset for the Twentieth Century (RGTracks-20C)

- 3
- 4 Guiling Ye<sup>1,2,3</sup>, Jeremy Cheuk-Hin Leung<sup>2,4\*</sup>, Wenjie Dong<sup>1,3\*</sup>, Jianjun Xu<sup>5</sup>, Weijing Li<sup>6</sup>, Weihong
- 5 Qian<sup>7</sup>, Hoiio Kong<sup>4</sup>, Banglin Zhang<sup>2,8</sup>

# 6 Affiliations

- 7 <sup>1</sup>School of Atmospheric Sciences, Key Laboratory of Tropical Atmosphere-Ocean System, Ministry of
- 8 Education, Sun Yat-sen University, Zhuhai, China
- 9 <sup>2</sup> Hunan Institute of Advanced Technology, Changsha, China
- 10 <sup>3</sup> Southern Marine Science and Engineering Guangdong Laboratory, Zhuhai, China
- 11 <sup>4</sup> Faculty of Data Science, City University of Macau, Macau, China
- 12 <sup>5</sup>CMA-GDOU Joint Laboratory for Marine Meteorology, South China Sea Institute of Marine
- 13 Meteorology, Guangdong Ocean University, Zhanjiang, China
- 14 <sup>6</sup>National Climate Center, China Meteorological Administration, Beijing, China
- 15 <sup>7</sup> Department of Atmospheric and Oceanic Sciences, Peking University, Beijing, China
- 16 <sup>8</sup> College of Atmospheric Science, Lanzhou University, Lanzhou, China
- 17
- 18
- 19
- 20 Corresponding author(s): Jeremy Cheuk-Hin Leung (chleung@pku.edu.cn); Wenjie Dong
- 21 (dongwj3@mail.sysu.edu.cn)

# 22 Abstract

| 23 | Tropical cyclones (TCs) are among the deadliest disasters affecting human society, and their            |
|----|---------------------------------------------------------------------------------------------------------|
| 24 | response to climate change has widely drawn attention from the public. However, assessing how           |
| 25 | historical TC activity changed with climate change has proven challenging due to incomplete TC          |
| 26 | records in the early years. Here, we introduce the Reanalysis-Based Global Tropical Cyclone Tracks      |
| 27 | Dataset for the Twentieth Century (RGTracks-20C) (Ye et al., 2024), a publicly available century-       |
| 28 | long global TC track dataset spanning from 1850–2014. The RGTracks-20C is reconstructed from            |
| 29 | the National Oceanic and Atmospheric Administration Twentieth Century Reanalysis using two              |
| 30 | independent TC tracking algorithms. Validation based on observations confirms that the RGTracks-        |
| 31 | 20C effectively captures the climatology and long-term variability of TC numbers, tracks, duration,     |
| 32 | and intensity across various ocean basins. A remarkable key strength of the RGTracks-20C is its         |
| 33 | ability to fill the missing intensity and location records of TCs observed in early years. This dataset |
| 34 | serves as a valuable historical data reference for future research on climate change and TC-related     |
| 35 | disasters.                                                                                              |
|    |                                                                                                         |

#### 36 1. Introduction

Tropical cyclones (TCs), also known as hurricanes or typhoons, are intense weather systems that 38 form over tropical and subtropical oceans and can cause severe disasters over the coastal regions and 39 even inland areas (Qin et al., 2024; Zhu and Quiring, 2022). Globally, approximately 80 TCs are 40 generated each year (Emanuel, 2018). As one of the most destructive weather systems (Bloemendaal et 41 al., 2022; Dinan, 2017; Emanuel, 2017), TCs significantly impact society and the economy (Kunze, 2021; 42 Lenzen et al., 2019; Noy, 2016). These impacts are expected to be exacerbated by climate change in the 43 future (Chan, 2023; Hassanzadeh et al., 2020; Knutson et al., 2020; Moon et al., 2023; Murakami and 44 Wang, 2022; Yamaguchi et al., 2020). Therefore, research on TCs has become increasingly vital in 45 climate change and prediction (Bhatia et al., 2019; Chan, 2019; Lanzante, 2019; Moon et al., 2019; 46 Sharmila and Walsh, 2018; Zhang et al., 2019). However, past variability of TC activity and underlying 47 mechanisms remains challenging due to incomplete early historical TC observation records, which may 48 lead to controversies (Chan et al., 2022a, b; Knutson et al., 2019; Lee et al., 2020).

Previous research has revealed significant issues related to the completeness of historical TC 50 observational data (Lee et al., 2020), which are highly dependent on the development of the global TC 51 observation system, data analysis techniques, and other factors (Klotzbach and Landsea, 2015; Knapp et 52 al., 2010; Kossin et al., 2020; Landsea et al., 2010; Mann et al., 2007; Ying et al., 2014). Before the 53 introduction of satellite observation, TC information (e.g., intensity and location) primarily relied on 54 conventional coastal weather stations and ship observation reports (Landsea et al., 2006, 2008). Aircraft 55 reconnaissance emerged in the North Atlantic (NATL) and western North Pacific (WNP) after World 56 War II (Emanuel, 2008). However, these observational techniques could not capture all occurred TCs 57 due to their limited observation range. It is possible that an existing TC was unrecorded in the early years. 58 In addition, even if a TC was observed and recorded, its track and intensity information may be 59 discontinuous due to the absence of meteorological satellite observations. For instance, there were no 60 observational records of TC wind speeds in the southern hemisphere before 1956 (Emanuel, 2021). Storm 61 intensity in the Indian Ocean is weaker compared to other basins, partly due to the lack of direct coverage 62 by geostationary satellites in that region before 1998 (Schreck et al., 2014). The incomplete observed 63 data of TCs in the early years, mainly before the 1970s, is a commonly-known unsolved issue in the 64 community.

Given the limitations of historical TC records, a promising approach is to utilize reanalysis for TC 66 identification (Li et al., 2024; Truchelut and Hart, 2011). Reanalysis combines historical observational 67 data with modern numerical weather models to produce comprehensive, continuous datasets of global 68 atmospheric conditions that adhere to physical principles (Compo et al., 2011; Kalnay et al., 1996; Parker, 69 2016; Slivinski, 2018). The Twentieth Century Reanalysis (20CR) (Compo et al., 2011), provided by the 70 National Oceanic and Atmospheric Administration (NOAA), is a global reanalysis dataset that covers the 71 longest period among all other reanalyses. The 20CR was designed for long-term analyses from 72 individual extreme weather events to climate variability, and has been applied to a wide range of studies, 73 including those on wave height, storm surge, Madden-Julian Oscillations, and TCs (Chand et al., 2022; 74 Cid et al., 2017; Gergis et al., 2020; Lee et al., 2023; Leung et al., 2022; Moore and Babij, 2017; Slivinski 75 et al., 2019; Truchelut et al., 2013; Wang et al., 2012). The fact that the 20CR only assimilates surface

pressure and sea level pressure fields, instead of other observations such as satellites and aircraft, makes 77 it less sensitive to the temporal inhomogeneity of observations (Slivinski et al., 2019, 2021). 78 Several independent studies have documented the feasibility of reproducing the characteristics of 79 some historical TC events based on the 20CR (Emanuel, 2010; Lee et al., 2023; Slivinski et al., 2019; 80 Truchelut et al., 2013; Truchelut and Hart, 2011). For example, following Emanuel (Emanuel, 2010), 81 who first expanded and revised TC climatology based on the 20CR, Truchelut and Hart (2011) employed 82 the 20CR to identify previously unknown TCs in the Atlantic and demonstrated that the 20CR can 83 accurately describe large-scale TC thermodynamic structure. Recently, Truchelut et al. (2013) noted that 84 the 20CR has the ability to investigate TC events that were previously undetected in the pre-satellite era. 85 Compared to other reanalyses, the 20CR well captures the intensity of the 1915 Galveston hurricane 86 (Slivinski et al., 2019) and also offers a more accurate representation of landfalling TCs in East Asia (Lee 87 et al., 2023). These previous studies have demonstrated the effectiveness of the 20CR as a tool for 88 characterizing historical TCs (Emanuel, 2010; Truchelut et al., 2013; Truchelut and Hart, 2011). Taking 89 advantage of the 20CR, some researchers have extracted the century-long TC information from the 90 reanalysis product (Chand et al., 2022; Lee et al., 2023; Yeasmin et al., 2023), suggesting its potential as 91 a tool for studying historical changes in TCs under anthropogenic climate change. 92 While the 20CR has been applied to studying the relationship between historical climate change and 93

TC variability, the primary focus was mostly on the TC occurrence frequency, and little attention was 94 given to other TC metrics such as intensity, duration, and location. More importantly, to date, there is no 95 publicly available reanalysis-based global TC dataset covering a century-long period. Therefore, the main 96 objective of this study is to extract TC information (including location, intensity, and lifetime) from the 97 20CR and reconstruct a historical global TC track dataset spanning 1850-2014. The produced dataset is 98 named the Reanalysis-Based Global Tropical Cyclone Tracks Dataset for the Twentieth Century 99 (RGTracks-20C) and is open to the public for research use. This paper first introduces the production 100 details of the RGTracks-20C and then discusses the validity, key strengths, and usage notes of the datasets. 101 We anticipate that the RGTracks-20C will provide valuable insights into the changing patterns of 102 historical TC activity, improving our understanding of the response of TCs to climate change.

#### 103 **2. Data and methods**

# 104 **2.1 Data**

105The primary objective of this study was to reconstruct a 20th century global TC dataset from the10620th Century Reanalysis version 3 (20CRv3) (Slivinski et al., 2019, 2021), the latest version of the 20CR107produced by NOAA. Then, the validity of the reconstructed 20th century global TC data was evaluated108based on the observed TC records, i.e., the International Best Track Archive for Climate Stewardship109(IBTrACS) dataset (Knapp et al., 2010).

# 110 2.1.1 20<sup>th</sup> Century Reanalysis

The 20CRv3 is led by NOAA's Physical Sciences Laboratory (PSL) and the Cooperative Institute

for Research in Environmental Sciences (CIRES) at the University of Colorado, supported by the U.S.

Department of Energy (DOE) (Slivinski et al., 2019, 2021). It, by combining advanced data assimilation

and numerical prediction techniques with historical observation data, provides long-term historical weather data with diverse variables, complete spatial and temporal coverage. The 20CRv3 employs seasurface temperature and sea-ice distributions as its boundary conditions and assimilates only surface pressure reports from the International Surface Pressure Databank (ISPD) version 4.7 (Compo et al., 2019; Cram et al., 2015), which include observations from stations and ships, as well as TC intensity (the minimum central pressure ( $SLP_{min}$ ) from the IBTrACS (Knapp et al., 2010). As such, it is more consistent and homogeneous with time than other reanalyses (Slivinski et al., 2019).

One should note that the IBTrACS and 20CRv3 are not two independent datasets because the SLP<sub>min</sub> 122 records in the IBTrACS are partly assimilated in the production of 20CRv3. However, reports show that 123 20CRv3 shows TCs structure and intensity more accurately and closer to observations than other 20th 124 century reanalyses as a result of the assimilation of IBTrACS (Laloyaux et al., 2018; Slivinski et al., 1252019). And, it provides a four-dimensional global gridded atmospheric dataset that spans the whole 20th 126 century and part of the 19th century (1836-2015, with an experimental extension spanning 1806-35), 127 with a 3-hour temporal resolution and  $1^{\circ} \times 1^{\circ}$  horizontal resolution (Slivinski et al., 2021). Thus, the 128 20CRv3 was applied to the production of the RGTracks-20C in this paper.

# 129 2.1.2 IBTrACS

The IBTrACS (Knapp et al., 2010), published by the NOAA, merges recent and historical TC data 131from meteorological agencies worldwide. These include the Regional Specialized Meteorological 132Centers (RSMC) and Tropical Cyclone Warning Centers (TCWC) of the World Meteorological 133 Organization (WMO), as well as non-WMO Centers, such as the China Meteorological Administration, 134 the Hong Kong Observatory and the Joint Typhoon Warning Center. The IBTrACS is the most 135comprehensive and publicly available global TC best-track dataset. It has been widely applied in previous 136 research to investigate the characteristics of TCs (Lai et al., 2020; Li et al., 2023; Tu et al., 2021, 2022; 137 Wang and Toumi, 2022; Zhang, 2023), and has served as a criterion for assessing TC records derived 138 from reanalysis (Bell et al., 2018; Bourdin et al., 2022; Chand et al., 2022; Hodges et al., 2017; Lee et al., 139 2023). In this study, the most updated version of IBTrACS (v04) (Knapp et al., 2018) serves as an 140 observation reference for evaluating the reliability of the RGTracks-20C. This dataset was cleaned before 141 being used for analyses. Details about the data pre-processing procedures are referred to in Figure B1 in 142Bourdin et al. (2022). In particular, we standardized maximum sustained wind speeds (WINDmax) in 143 IBTrACS to 10-minute sustained wind speeds to ensure a consistent global standard(Knapp et al., 2010). 144 We then removed tracks that did not reach the tropical storm stage ( $WIND_{max}

# 152 **2.2 Production of the RGTracks-20C**

#### 153 **2.2.1 Procedure**

154The RGTracks-20C was constructed from the latest version of 20CR (20CRv3). The relatively short 155and imperfectly sampled observational record of TCs introduces considerable uncertainty in their data 156over the past century (Landsea, 2007; Landsea et al., 2010), hindering accurate detection of interannual 157variability and long-term trends (Knutson et al., 2019; Lee et al., 2020). Reanalysis is an effective way 158to reduce this uncertainty (Chand et al., 2022; Truchelut et al., 2013). Since TC information is not directly 159provided in the 20CRv3, objective TC trackers were applied to detect and track TCs in this dataset. 160 Numerous trackers have been developed by operational centers and research institutions to meet various 161 application needs (Hodges et al., 2017; Horn et al., 2014; Tory et al., 2013; Zarzycki and Ullrich, 2017). 162 In this study, as the first version of the RGTracks-20C, we applied two widely used, publicly available, 163 and effective trackers: (1) the physically-based Ullrich & Zarzycki (UZ) tracker (Zarzycki and Ullrich, 164 2017) and (2) the dynamics-based Okubo-Weiss-Zeta (OWZ) tracker (Tory et al., 2013). Both trackers 165 have been reported to effectively capture TC systems from coarse resolution gridded data uncertainty 166 (Chand et al., 2022; Truchelut et al., 2013), such as the 20CRv3. Figure 1 shows the procedure of 167 producing the RGTracks-20C, and details of the methodology are provided in the following.