# Peer review of "A Reanalysis-Based Global Tropical Cyclone Tracks"

_Earth System Science Data, 2025_

## Author Comment (AC2)

**Review 2**

A Reanalysis-Based Global Tropical Cyclone Tracks Dataset for the Twentieth Century (RGTracks-20C)

**General Statement**:

The manuscript by Ye and the colleagues presents a reconstructed tropical cyclone (TC) dataset using two commonly applied tracker algorithms, OWZ and UZ. The authors evaluate the climatology and long-term variability of TC frequency, track patterns, duration, and intensity across various ocean basins, comparing their RGTracks-20C product with the IBTrACS dataset. They argue that a key advantage of RGTracks-20C lies in its ability to fill gaps in historical TC intensity and location records, particularly in the earlier part of the record. However, considering the lack of methodological novelty and the limitations associated with the underlying reanalysis dataset, I am not convinced that this study meets the publication standards of ESSD.

**Authors' response:** Thank you very much for your valuable comments and constructive suggestions on our manuscript. You have raised two important points regarding our work: the novelty of employed approaches and the choice of reanalysis datasets used for historical TC reconstruction. We appreciate your careful consideration of our manuscript. In the following, we attempt to respond to and address your concerns.

**Comments:**

**Pont 1.** The authors appear to apply two widely used trackers (OWZ and UZ) without any evident modification or improvement. Notably, the UZ tracker has recently been enhanced through the integration of AI-based techniques, showing considerable performance gains over its original version (Han & Ullrich, 2025). A new TC dataset has been reconstructed using this improved approach. If the authors have indeed introduced any innovation or adaptation to the tracking algorithms, this should be clearly detailed in the manuscript. Otherwise, the study amounts to a straightforward application of existing (and arguably outdated) methods to the 20CRv3 dataset. From this perspective, the contribution seems largely computational and lacks substantive novelty.

**Point 2**. The limitations of the 20CRv3 reanalysis dataset have been discussed in detail by Emanuel (2024). Moreover, previous studies have already generated TC track datasets using reanalysis data, such as ERA5, in conjunction with various tracking algorithms (e.g., Bourdin et al., 2022). A comparative assessment of TC tracks derived from different reanalysis datasets (e.g., ERA5 vs. 20CRv3) using identical or similar trackers would have added substantial value by helping validate the reliability of each dataset and identify the more suitable one for historical TC reconstruction.

**Authors' response:** In the following, we would like to address your concerns regarding the methodological novelty and alignment with Earth System Science Data (ESSD) publication standards.

(1) We respectfully acknowledge that our work does not introduce novel tracking algorithms. The two tracking algorithms employed in our study, the UZ and OWZ, are respectively proposed by Zarzycki and Ullrich (Ullrich et al., 2021; Zarzycki and Ullrich, 2017) and Tory et al. (2013). These two algorithms are widely recognized and publicly available, and have been shown in previous studies to be well-suited for long-term tracking of TCs in reanalyses (Bell et al., 2018; Bourdin et al., 2022; Chand et al., 2022; Ullrich et al., 2021). Studies such as Bourdin et al. (2022) have demonstrated the efficiency and cost-effectiveness of these algorithms, particularly for large datasets like 20CRv3.

**However, we would like to emphasize that the primary focus of this study is not on testing or developing new TC tracking algorithms. The primary goal of this study, as well as that of the RGTracks-20C project, is to develop a century-long temporal span (starting from the mid-nineteenth century) and a publicly accessible global tropical cyclone track dataset, where little to none had existed before (as commented by Reviewer #1).** The first version of RGTracks-20C, introduced in our submitted manuscript, effectively fills critical gaps in tropical cyclone observations prior to the satellite era, particularly regarding intensity and track information and previously unrecorded tropical cyclone events. We believe that the RGTracks-20C dataset is valuable historical data that could aid future research on the response of tropical cyclone activity to historical anthropogenic climate change.

According to ESSD's official scope, the journal "*encourages submissions on original data or data collections which are of sufficient quality and have the potential to contribute to Earth*

*system sciences*" (Carlson and Oda, 2018). Although we did not introduce novel tracking algorithms in this manuscript, the RGTracks-20C dataset is, to our knowledge, the very first publicly available historical global tropical cyclone track dataset covering the whole 20th century.

The contribution of our work lies not in algorithmic novelty, but in its (1) century-long time span (1850–2014), (2) ability to fill critical gaps in tropical cyclone track and intensity observations prior to the satellite era. In addition, the RGTracks-20C also promotes the development and application of 20CRv3 reanalysis data in tropical cyclone research while providing a benchmark for quality assessment of other related data products. Given the above contributions, the RGTracks-20C can serve as an important scientific foundation and technical reference for research institutions and operational agencies in revising historical tropical cyclone datasets. By analyzing historically impacting but unrecorded tropical cyclone events, we can quantify their associated storm surge, wind damage, and precipitation impacts, etc., providing invaluable scientific support for disaster risk reduction planning in currently vulnerable coastal cities. Thus, we strongly believe that our work aligns perfectly with the mission of ESSD journal, which is dedicated to promoting open access to high-quality datasets that have the potential to contribute to research on Earth System Science.

(2) The second concern you raised is about the reliability or the limitations of the 20CRv3 reanalysis dataset. Here, we would like to note again that the goal of this study is to develop a century-long temporal span and publicly accessible global tropical cyclone track dataset. This could only be achieved by employing 20th century reanalyses, but not ERA5 or other reanalysis datasets that are produced based on satellite data. This is the primary reason why the 20CRv3, but not ERA5, was employed in our study. In addition, the ability of 20CRv3 to capture tropical cyclone information has been confirmed in previous studies. Allow us to explain in more detail below.

In the context of global climate change, there have been significant changes in the frequency and intensity of extreme weather events (Grant et al., 2025), especially cyclones (Knutson et al., 2010). This type of weather system often causes severe casualties and economic losses (Young and Hsiang, 2024), making a deeper understanding of their formation mechanisms and long-term variations crucial for improving forecasting capabilities. However,

due to limitations in observational data (Chan et al., 2022; Lanzante, 2019; Torn and Snyder, 2012), most of the existing studies have focused on tropical cyclone activities since 1980 (Bhatia et al., 2019; Yamaguchi et al., 2020), which has severely constrained the understanding of the characteristics of long-term tropical cyclone activities and their climate responses. To break through this limitation, it is necessary to extend the length and improve the quality of the historical tropical cyclone databases.

In this aspect of reconstructing historical tropical cyclone datasets, one of the signature works is the Atlantic basin hurricane database (HURDAT) reevaluation project (Landsea et al., 2004, 2008), which provides the estimated positions and intensities of all tropical storms, subtropical storms and hurricanes recorded in the Atlantic Basin since 1851. Also, a team of scientists has reconstructed data of tropical cyclone landfalls in Japan over a 142-year period (1877-2019) (Kubota et al., 2021). And, Cerrito (2018) has reconstructed the tropical cyclone tracks and intensity of the first three hurricanes that occurred before 1850 by integrating various sources of data such as ship logs, newspapers, and instrument records, and combining Geographic Information System (GIS) technology (Cerrito, 2018). This type of tropical cyclone data reconstruction work mainly relies on traditional materials such as ship records and historical reports, and its uncertainty is relatively high (Lanzante, 2019; Vecchi et al., 2021).

To address this issue, Emanuel (2010) pioneered the exploration of the potential of using reanalysis data for revising historical TC activity. Reanalysis data, characterized by their extended temporal span and complete spatial coverage, are of significant value in the study of extreme weather events, particularly tropical cyclones, and have been widely applied (Chand et al., 2022; Lee et al., 2023; Yeasmin et al., 2023). Truchelut and Hart (2011) initially employed manual methods to identify TC candidate events from the Twentieth-Century Reanalysis (20CR) data that were not recorded in best track observation datasets. These verified candidate events were then submitted to the U.S. National Hurricane Center (NHC) to support the revision of HURDAT. Subsequently, efficient and objective detection methods were further developed specifically for identifying TC candidate events in the pre-satellite era, with the aim of assisting current and future teams involved in the revision of climatology by providing a systematic dataset of candidate events (Truchelut et al., 2013). Since then, studies on tropical cyclones based on reanalysis data have been increasing, but most of these studies are still limited to the

period after the mid-20th century (Bourdin et al., 2022; Han and Ullrich, 2025; Li et al., 2024; Xu et al., 2024).

Among the existing reanalysis datasets, the 20CR dataset offers the most extensive temporal coverage, extending back to the 19th century. Previous studies have indicated that 20CR data can effectively validate the authenticity of historical TC records (Lee et al., 2023; Truchelut et al., 2013; Truchelut and Hart, 2011; Yeasmin et al., 2023). For instance, Chand et al. (2022) and Yeasmin et al. (2023) utilized 20CRv2c for TC activity reconstruction, thereby verifying the reliability of 20CR data in such research. Compared to its earlier versions, the 20CRv3 exhibits significant improvements in spatial resolution, the accuracy of storm intensity representation, and error control, along with a substantial reduction in model systematic biases (Slivinski et al., 2019). In addition, comparative analyses show that the performance of RGTracks-20C after 1980 is consistent with the current ERA5 reanalysis with high resolution and best quality (Lee et al., 2023). Our validation analysis also yields consistent results (see Supplementary Section S1 and Table S1).

Taking the above factors into consideration, while we agree that the overall quality of 20CRv3 may not be as good as ERA5, we believe that 20CRv3, which provides data records from the mid-nineteenth century, is an ideal data source for constructing the first version of the RGTracks-20C dataset.

(3) You have also mentioned the most updated UZ tracker, which has recently been enhanced through the integration of the Systematic Cyclone Low-Pressure System (SyCLoPS) unified objective framework (Han and Ullrich, 2025). We would like to thank you for your valuable suggestions. We are currently attempting to implement Han's algorithm and have successfully configured partial components of the system.

In future work, we plan to apply machine learning-based approaches to 20CRv3 reanalysis data, incorporating advanced algorithms such as the SyCLoPS framework (Han and Ullrich, 2025) and its automated classification methodology to improve both computational efficiency and detection accuracy for low-pressure system identification and tracking.

**References**

Bell, S. S., Chand, S. S., Tory, K. J., and Turville, C.: Statistical Assessment of the OWZ Tropical Cyclone Tracking Scheme in ERA-Interim, Journal of Climate, 31, 2217–2232, https://doi.org/10.1175/JCLI-D-17-0548.1, 2018.

Bhatia, K. T., Vecchi, G. A., Knutson, T. R., Murakami, H., Kossin, J., Dixon, K. W., and Whitlock, C. E.: Recent increases in tropical cyclone intensification rates, Nat Commun, 10, 635, https://doi.org/10.1038/s41467-019-08471-z, 2019.

Bourdin, S., Fromang, S., Dulac, W., Cattiaux, J., and Chauvin, F.: Intercomparison of four algorithms for detecting tropical cyclones using ERA5, Geoscientific Model Development, 15, 6759–6786, https://doi.org/10.5194/gmd-15-6759-2022, 2022.

Carlson, D. and Oda, T.: Editorial: Data publication – *ESSD* goals, practices and recommendations, Earth System Science Data, 10, 2275–2278, https://doi.org/10.5194/essd-10-2275-2018, 2018.

Cerrito, E.: Reconstructing Historical Hurricane Tracks in the Atlantic Basin: Three Case Studies from the 1840s, USF Tampa Graduate Theses and Dissertations, 2018.

Chan, K. T. F., Chan, J. C. L., Zhang, K., and Wu, Y.: Uncertainties in tropical cyclone landfall decay, npj Clim Atmos Sci, 5, 1–8, https://doi.org/10.1038/s41612-022-00320-z, 2022.

Chand, S. S., Walsh, K. J. E., Camargo, S. J., Kossin, J. P., Tory, K. J., Wehner, M. F., Chan, J. C. L., Klotzbach, P. J., Dowdy, A. J., Bell, S. S., Ramsay, H. A., and Murakami, H.: Declining tropical cyclone frequency under global warming, Nat. Clim. Chang., 12, 655–661, https://doi.org/10.1038/s41558-022-01388-4, 2022.

Emanuel, K.: Tropical Cyclone Activity Downscaled from NOAA-CIRES Reanalysis, 1908–1958, Journal of Advances in Modeling Earth Systems, 2, https://doi.org/10.3894/JAMES.2010.2.1, 2010.

Grant, L., Vanderkelen, I., Gudmundsson, L., Fischer, E., Seneviratne, S. I., and Thiery, W.: Global emergence of unprecedented lifetime exposure to climate extremes, Nature, 641, 374–379, https://doi.org/10.1038/s41586-025-08907-1, 2025.

Han, Y. and Ullrich, P. A.: The System for Classification of Low-Pressure Systems (SyCLoPS): An All-In-One Objective Framework for Large-Scale Data Sets, Journal of Geophysical

Research: Atmospheres, 130, e2024JD041287, https://doi.org/10.1029/2024JD041287, 2025.

Knutson, T. R., McBride, J. L., Chan, J., Emanuel, K., Holland, G., Landsea, C., Held, I., Kossin, J. P., Srivastava, A. K., and Sugi, M.: Tropical cyclones and climate change, Nature Geosci, 3, 157–163, https://doi.org/10.1038/ngeo779, 2010.

Kubota, H., Matsumoto, J., Zaiki, M., Tsukahara, T., Mikami, T., Allan, R., Wilkinson, C., Wilkinson, S., Wood, K., and Mollan, M.: Tropical cyclones over the western north Pacific since the mid-nineteenth century, Climatic Change, 164, 29, https://doi.org/10.1007/s10584-021-02984-7, 2021.

Landsea, C. W., Franklin, J. L., McAdie, C. J., Beven, J. L., Gross, J. M., Jarvinen, B. R., Pasch, R. J., Rappaport, E. N., Dunion, J. P., and Dodge, P. P.: A Reanalysis of Hurricane Andrew's Intensity, Bulletin of the American Meteorological Society, 85, 1699–1712, https://doi.org/10.1175/BAMS-85-11-1699, 2004.

Landsea, C. W., Glenn, D. A., Bredemeyer, W., Chenoweth, M., Ellis, R., Gamache, J., Hufstetler, L., Mock, C., Perez, R., Prieto, R., Sánchez-Sesma, J., Thomas, D., and Woolcock, L.: A Reanalysis of the 1911–20 Atlantic Hurricane Database, Journal of Climate, 21, 2138–2168, https://doi.org/10.1175/2007JCLI1119.1, 2008.

Lanzante, J. R.: Uncertainties in tropical-cyclone translation speed, Nature, 570, E6–E15, https://doi.org/10.1038/s41586-019-1223-2, 2019.

Lee, R., Chen, L., and Ren, G.: A comparison of East-Asia landfall tropical cyclone in recent reanalysis datasets--before and after satellite era, Frontiers in Earth Science, 10, 2023.

Li, J., Tian, Q., Shen, Z., Xu, Y., Yan, Z., Li, M., Zhu, C., Xue, J., Lin, Z., Yang, Y., and Zeng, L.: Fidelity of global tropical cyclone activity in a new reanalysis dataset (CRA40), Meteorological Applications, 31, e70009, https://doi.org/10.1002/met.70009, 2024.

Slivinski, L. C., Compo, G. P., Whitaker, J. S., Sardeshmukh, P. D., Giese, B. S., McColl, C., Allan, R., Yin, X., Vose, R., Titchner, H., Kennedy, J., Spencer, L. J., Ashcroft, L., Brönnimann, S., Brunet, M., Camuffo, D., Cornes, R., Cram, T. A., Crouthamel, R., Domínguez-Castro, F., Freeman, J. E., Gergis, J., Hawkins, E., Jones, P. D., Jourdain, S., Kaplan, A., Kubota, H., Blancq, F. L., Lee, T.-C., Lorrey, A., Luterbacher, J., Maugeri, M., Mock, C. J., Moore, G. W. K., Przybylak, R., Pudmenzky, C., Reason, C., Slonosky, V. C.,

Smith, C. A., Tinz, B., Trewin, B., Valente, M. A., Wang, X. L., Wilkinson, C., Wood, K., and Wyszyński, P.: Towards a more reliable historical reanalysis: Improvements for version 3 of the Twentieth Century Reanalysis system, Quarterly Journal of the Royal Meteorological Society, 145, 2876–2908, https://doi.org/10.1002/qj.3598, 2019.

Torn, R. D. and Snyder, C.: Uncertainty of Tropical Cyclone Best-Track Information, https://doi.org/10.1175/WAF-D-11-00085.1, 2012.

Tory, K. J., Chand, S. S., McBride, J. L., Ye, H., and Dare, R. A.: Projected Changes in Late-Twenty-First-Century Tropical Cyclone Frequency in 13 Coupled Climate Models from Phase 5 of the Coupled Model Intercomparison Project, Journal of Climate, 26, 9946–9959, https://doi.org/10.1175/JCLI-D-13-00010.1, 2013.

Truchelut, R. E. and Hart, R. E.: Quantifying the possible existence of undocumented Atlantic warm-core cyclones in NOAA/CIRES 20th Century Reanalysis data, Geophysical Research Letters, 38, https://doi.org/10.1029/2011GL046756, 2011.

Truchelut, R. E., Hart, R. E., and Luthman, B.: Global Identification of Previously Undetected Pre-Satellite-Era Tropical Cyclone Candidates in NOAA/CIRES Twentieth-Century Reanalysis Data, Journal of Applied Meteorology and Climatology, 52, 2243–2259, https://doi.org/10.1175/JAMC-D-12-0276.1, 2013.

Ullrich, P. A., Zarzycki, C. M., McClenny, E. E., Pinheiro, M. C., Stansfield, A. M., and Reed, K. A.: TempestExtremes v2.1: a community framework for feature detection, tracking, and analysis in large datasets, Geoscientific Model Development, 14, 5023–5048, https://doi.org/10.5194/gmd-14-5023-2021, 2021.

Vecchi, G. A., Landsea, C., Zhang, W., Villarini, G., and Knutson, T.: Changes in Atlantic major hurricane frequency since the late-19th century, Nat Commun, 12, 4054, https://doi.org/10.1038/s41467-021-24268-5, 2021.

Xu, Z., Guo, J., Zhang, G., Ye, Y., Zhao, H., and Chen, H.: Global tropical cyclone size and intensity reconstruction dataset for 1959–2022 based on IBTrACS and ERA5 data, Earth System Science Data, 16, 5753–5766, https://doi.org/10.5194/essd-16-5753-2024, 2024.

Yamaguchi, M., Chan, J. C. L., Moon, I.-J., Yoshida, K., and Mizuta, R.: Global warming changes tropical cyclone translation speed, Nat Commun, 11, 47, https://doi.org/10.1038/s41467-019-13902-y, 2020.

Yeasmin, A., Chand, S., and Sultanova, N.: Reconstruction of tropical cyclone and depression proxies for the South Pacific since the 1850s, Weather and Climate Extremes, 39, 100543, https://doi.org/10.1016/j.wace.2022.100543, 2023.

Young, R. and Hsiang, S.: Mortality caused by tropical cyclones in the United States, Nature, 635, 121–128, https://doi.org/10.1038/s41586-024-07945-5, 2024.

Zarzycki, C. M. and Ullrich, P. A.: Assessing sensitivities in algorithmic detection of tropical cyclones in climate data, Geophysical Research Letters, 44, 1141–1149, https://doi.org/10.1002/2016GL071606, 2017.

---

## Author Comment (AC3)

We sincerely appreciate your comprehensive evaluation and constructive feedback on our study. We note your concern regarding the novelty of the research methodology. Thus, in this follow-up response, we have implemented the System for Classification of Low-Pressure Systems (SyCLoPS) algorithm developed by Han and Ullrich (2025), as suggested by you, to our 20CRv3 dataset for comparative evaluation.

In your review report, you mentioned that "*The authors appear to apply two widely used trackers without any evident modification, notably when the UZ tracker has recently been enhance*". Following your advice, we have applied the SyCLoPS algorithm to extract tropical cyclone information from the 20CRv3. Then, we conducted an evaluation on the results of 2010 to compare the performance of different tracking algorithms (Fig. R1 and R2). Our analysis reveals that the OWZ algorithm (Tory et al., 2013) maintains the highest POD, followed by SyCLoPS (Han and Ullrich, 2025), and then UZ (Ullrich et al., 2021). In terms of the number of successful tropical cyclone identifications, OWZ achieved 51, followed by SyCLoPS with 47 and UZ with 43. However, when considering FAR, SyCLoPS demonstrates superior performance with only 6%, significantly lower than UZ (14%) and OWZ (26%). This indicates that while OWZ achieves the highest detection rate, it also produces more false positives. SyCLoPS offers a balanced approach with relatively high detection capability and notably low false alarm rates.

Notably, when compared to other AI-based algorithms (Accarino et al., 2023) that identify tropical cyclones from ERA5 data (Table. R1 green), the UZ and OWZ algorithms we applied to 20CRv3 also demonstrate competitive performance. Based on these quantitative results, we conclude that while algorithmic refinements contribute to improved performance, the impact on overall detection capabilities remains relatively modest. The UZ and OWZ algorithms currently employed in our study demonstrate satisfactory reliability and accuracy for our research objectives.

[Figure]

**Figure R1: Accuracy of different TC tracking algorithms in identifying TCs in 2010. a–b, POD (blue bars and line, unit: %) and FAR (red bars and line, unit: %) for TC number detected by the UZ (A), OWZ (B) and SyCLoPS (C) trackers in each basin (bars), compared to the global mean (lines). Blue and red horizontal lines denote the POD and FAR over the globe. D–F, same as C–D, except for the number of hits (blue bars), misses (green bars), and false alarms (red bars) detected by the UZ (D), OWZ (E) and SyCLoPS (F) trackers.**

[Figure]

**Figure R2: TC genesis locations (red dots) and tracks (blue lines) from IBTrACS (A) and tracking algorithms: UZ (B), and OWZ (C) and SyCLoPS (D) in 2010.**

**Table R1: The probability of detection (POD) and false alarm rate (FAR) of the global TCs detected by different trackers in the fifth generation ECMWF reanalysis (ERA5) and 20CRv3.** POD (unit: %) and FAR (unit: %) of TCs detected by different trackers in the latest high-resolution ERA5 reanalysis by (Accarino et al., 2023) (green shading), (Bourdin et al., 2022) (blue shading), and RGTracks-20C (orange shading).

|  | Hybrid | CNRM | TRACK | UZ-ERA5 | OWZ-ERA5 | UZ-20CRv3 | OWZ-20CRv3 |
|---|---|---|---|---|---|---|---|
| **POD (%)** | 71.49 | 72.77 | 74.37 | 71.54 | 71.75 | 67.62 | 76.56 |
| **FAR (%)** | 23 | 8.62 | 17.19 | 3.37 | 17.43 | 7.19 | 15.21 |

To further validate the algorithm performance, we selected Typhoon "MEGI" that occurred in the Western North Pacific during 2010 for detailed analysis. All three algorithms successfully detected this typhoon from the 20CRv3 and accurately

reproduced its observed track (Fig. R3). Notably, SyCLoPS provided stage classification throughout the typhoon's lifespan, with the tropical cyclone genesis and dissipation phases showing good agreement with IBTrACS observations (Fig. R4). This finding is consistent with the results reported by Han and Ullrich (2025).

We acknowledge the significant advantages of the SyCLoPS algorithm, particularly its capability to classify different developmental stages and cyclone types throughout their complete lifespans. This functionality provides invaluable information for studying tropical cyclone evolution and stage-specific characteristics that is not available through the OWZ and UZ algorithms alone. The stage classification feature represents a substantial advancement in cyclone tracking methodology, offering enhanced analytical capabilities for understanding cyclone dynamics and morphology.

Recognizing these benefits, we have initiated the application of the SyCLoPS methodology to the complete 20CRv3 dataset to supplement the RGTracks-20C database with tropical cyclone classification information. This ongoing work aims to provide the scientific community with enhanced cyclone stage categorization capabilities for more detailed climatological and dynamical studies.

Currently, we have already obtained preliminary results from our 2010 test dataset for reference and validation purposes. This additional classification information provides valuable supplementary data that complements the existing OWZ and UZ algorithm outputs. We plan to incorporate these enhanced classification features in future versions of the RGTracks-20C dataset, pending completion of the full dataset processing. Thank you once again for your valuable suggestion that help improve our dataset a lot.

[Figure]

**Figure R3: Best track comparison of Typhoon "MEGI" (2010) from IBTrACS and tracking algorithms: IBTrACS (blue), SyCLoPS (orange), UZ (green), and OWZ (red).**

[Figure]

**Figure R4: Classification of different low-pressure system stages during the lifetime of Typhoon "MEGI" (2010) as identified by the SyCLoPS algorithm. TLO indicates Tropical Low, TD indicates Tropical Depression, and TC indicates Tropical Cyclone. The cross marks indicate the position of the IBTrACS record start (black), the first IBTrACS tropical cyclone classification (red), and the IBTrACS record end (gray).**

**References**

Accarino, G., Donno, D., Immorlano, F., Elia, D., and Aloisio, G.: An Ensemble Machine Learning Approach for Tropical Cyclone Localization and Tracking From ERA5 Reanalysis Data, Earth Space Sci., 10, e2023EA003106, https://doi.org/10.1029/2023EA003106, 2023.

Bourdin, S., Fromang, S., Dulac, W., Cattiaux, J., and Chauvin, F.: Intercomparison of four algorithms for detecting tropical cyclones using ERA5, Geosci. Model Dev., 15, 6759–6786, https://doi.org/10.5194/gmd-15-6759-2022, 2022.

Han, Y. and Ullrich, P. A.: The System for Classification of Low-Pressure Systems (SyCLoPS): An All-In-One Objective Framework for Large-Scale Data Sets, J. Geophys. Res. Atmospheres, 130, e2024JD041287, https://doi.org/10.1029/2024JD041287, 2025.

Tory, K. J., Dare, R. A., Davidson, N. E., McBride, J. L., and Chand, S. S.: The importance of low-deformation vorticity in tropical cyclone formation, Atmospheric Chem. Phys., 13, 2115–2132, https://doi.org/10.5194/acp-13-2115-2013, 2013.

Ullrich, P. A., Zarzycki, C. M., McClenny, E. E., Pinheiro, M. C., Stansfield, A. M., and Reed, K. A.: TempestExtremes v2.1: a community framework for feature detection, tracking, and analysis in large datasets, Geosci. Model Dev., 14, 5023–5048, https://doi.org/10.5194/gmd-14-5023-2021, 2021.